# Disrupting Diffusion: Token-Level Attention Erasure Attack against Diffusion-based Customization

### Yisu Liu
Institute of Information Engineering,
Chinese Academy of Sciences &
School of Cyber Security, University
of Chinese Academy of Sciences
Beijing, China
liuyisu@iie.ac.cn

### Jinyang An
Institute of Information Engineering,
Chinese Academy of Sciences &
School of Cyber Security, University
of Chinese Academy of Sciences
Beijing, China
anjinyang@iie.ac.cn

### Wanqian Zhang*
Institute of Information Engineering,
Chinese Academy of Sciences
Beijing, China
zhangwanqian@iie.ac.cn

### Dayan Wu
Institute of Information Engineering,
Chinese Academy of Sciences
Beijing, China
wudayan@iie.ac.cn

### Jingzi Gu
Institute of Information Engineering,
Chinese Academy of Sciences
Beijing, China
gujingzi@iie.ac.cn

### Zheng Lin
Institute of Information Engineering,
Chinese Academy of Sciences &
School of Cyber Security, University
of Chinese Academy of Sciences
Beijing, China
linzheng@iie.ac.cn

### Weiping Wang
Institute of Information Engineering,
Chinese Academy of Sciences &
School of Cyber Security, University
of Chinese Academy of Sciences
Beijing, China
wangweiping@iie.ac.cn

## Abstract

With the development of diffusion-based customization methods like DreamBooth, individuals now have access to train the models that can generate their personalized images. Despite the convenience, malicious users have misused these techniques to create fake images, thereby triggering a privacy security crisis. In light of this, proactive adversarial attacks are proposed to protect users against customization. The adversarial examples are trained to distort the customization model's outputs and thus block the misuse. In this paper, we propose DisDiff (Disrupting Diffusion), a novel adversarial attack method to disrupt the diffusion model outputs. We first delve into the intrinsic image-text relationships, well-known as cross-attention, and empirically find that the subject-identifier token plays an important role in guiding image generation. Thus, we propose the Cross-Attention Erasure module to explicitly "erase" the indicated attention maps and disrupt the text guidance. Besides, we analyze the influence of the sampling process of the diffusion model on Projected Gradient Descent (PGD) attack and introduce a novel Merit Sampling Scheduler to adaptively modulate the perturbation updating amplitude in a step-aware manner. Our DisDiff outperforms the state-of-the-art methods by 12.75% of FDFR scores and 7.25% of ISM scores across two facial benchmarks and two commonly used prompts on average.

## CCS Concepts

• **Computing methodologies → Computer vision**.

## Keywords

Diffusion models, adversarial attack, anti-customization

### ACM Reference Format:
Yisu Liu, Jinyang An, Wanqian Zhang, Dayan Wu, Jingzi Gu, Zheng Lin, and Weiping Wang. 2024. Disrupting Diffusion: Token-Level Attention Erasure Attack against Diffusion-based Customization. In *Proceedings of the 32nd ACM International Conference on Multimedia (MM '24), October 28-November 1, 2024, Melbourne, VIC, Australia.* ACM, New York, NY, USA, 10 pages. https://doi.org/10.1145/3664647.3681243

*indicates corresponding author.

## 1 Introduction

In the era of pre-trained generative AI, individuals wield the power of creating multi-media contents that appears genuinely authentic with just a single click. Despite the success of GAN- and VAE-based image synthesis methods[19, 25, 29, 33, 41, 49], more impressive and controllable results are achieved by recently proposed diffusion models, such as DALL-E[32], Imagen[38] and Stable Diffusion[40]. These diffusion model based text-to-image generation methods can

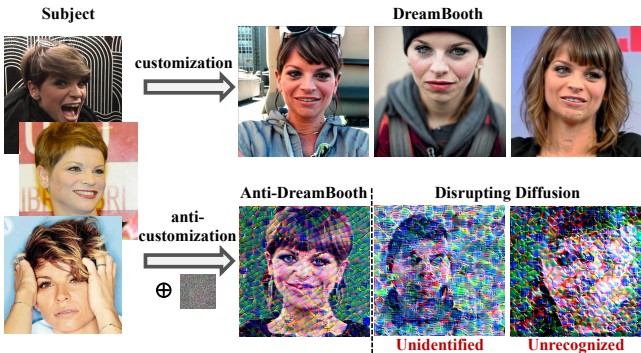

**Figure 1: Comparison of DreamBooth, Anti-DreamBooth with DisDiff. The first row shows the unprotected Dream-Booth, which learns from the subjects and generates them into different scenes. The second row shows the protected results. Protected by Anti-DreamBooth, the face is still identi-fied. Our DisDiff achieves unidentified or even unrecognized outputs, boosting the protection performances.**

generate almost any style of image by a simple corresponding text instruction, and show the unprecedented capacity for producing high-quality images across various applications.

With the users' desire for image synthesis of their subjects, e.g., daily supplies, pets or family members, customized image generation steps onto the stage, with representatives like Textual Inversion[11] and DreamBooth[37]. Leveraging the pre-trained text-to-image diffusion models, Textual Inversion learns a pseudo-word in the well-editable word embedding space of text encoder, i.e., $S*$, to represent the newly provided concept. DreamBooth fine-tunes the diffusion models with a small set of subject images and a subject-identifier token (e.g., *sks*) included prompt, such as "a photo of *sks* dog". Iteratively training large amounts of model parameters, DreamBooth achieves superior performance compared to Textual Inversion. More recently, Custom Diffusion[21] fine-tunes the se-lected parameters in the attention layers to reduce the memory overhead, achieving efficient customization.

Despite the convenient editing of users' specified subjects, these methods present a dual-edged dilemma. Malicious users exploit this customization to fabricate synthetic images or videos and spread them on the internet. The generated fake products[6, 20, 51] en-gender potential security and privacy vulnerabilities for individu-als. In response to defend against this, both passive and proactive protections have been proposed for safeguarding privacy. Passive protections, commonly known as DeepFake detection[12, 18, 24, 46, 53, 54], are employed to determine the authenticity of images. The detecting models are trained to discriminate whether the im-ages are real or fake. However, fake images have been spread be-fore detected and these methods often struggle with novel syn-thetic techniques[45]. Regarding this matter, proactive protections are proposed with adversarial attacks to disrupt the fake image synthesis[48, 50]. These applications distort the fake image genera-tion with noticeable artifacts, thereby providing cues for detection.

As diffusion-based customization methods emerge, the urgency of protecting users' privacy against these techniques becomes para-mount. As one of the pioneering works, Anti-DreamBooth[42] aims to disrupt the generation of images produced by DreamBooth. Specifically, Anti-DreamBooth tries to alternatively learn a sur-rogate diffusion model (or utilize a fine-tuned fixed one) and the adversarial perturbations to improve the attack performances. De-spite the destruction of the visual quality of generated images by Anti-DreamBooth, ***the face still exists***. This is obvious not only for human observers (visually) but also for detection algorithms (quantitatively). As illustrated in Figure 1, the generated images of both DreamBooth and Anti-DreamBooth are recognized, i.e., there exists an evident face in the image. Besides, the generated face can even be correctly identified, i.e., being classified as the same identity by a face recognition algorithm. These undesired issues of privacy information leakage become the motivation of our work.

In this paper, we propose DisDiff, a novel adversarial attack method against diffusion-based customization. We first delve into the training process of DreamBooth through the lens of the atten-tion mechanism. As training proceeds, the diffusion model gradually establishes the association between "*sks* person" and the target indi-vidual. This association can be reflected by the cross-attention maps in the UNet[35], which serves as the core guidance of text-image generation. In light of this, we propose a novel Cross-Attention Era-sure module (CAE), to diminish the cross-attention's guiding effects. On the other hand, we explore the impact of the sampling proce-dure of diffusion models on adversarial attacks inspired by [44]. We compute Hybrid Quality Scores (HQS) for different timesteps to measure the importance of these steps on adversarial learning. Then, we introduce the Merit Sampling Scheduler (MSS) to adaptively modulate the PGD step sizes during attack, i.e., at the steps where HQS is large, PGD step size is magnified by MSS and vice versa. Both CAE and MSS modules are empirically proven to boost the adversarial attack performances on diffusion-based customizations.

In brief, our contributions are summarised as follows:

1) We propose DisDiff, a proactive adversarial attack method against diffusion-based customization, to protect users' privacy.

2) We delve into the image-text relationship and justify its im-portance on generation guidance. On this basis, we propose a novel Cross-Attention Erasure module to "erase" the subject-identifier token-related attention map.

3) Observing the diffusion sampling process and PGD, we intro-duce a Merit Sampling Scheduler, which restricts the perturbation updating step length in a step-aware manner.

4) Experiments show that DisDiff outperforms the SOTA method by 12.75% of FDFR scores and 7.25% of ISM scores across two facial benchmarks and commonly used prompts on average.

## 2 RELATED WORK

### 2.1 Text-to-image Diffusion

As latent-variable generative models, diffusion models establish a Markov chain of continuous timesteps to gradually introduce Gaussian noise into the training data. Afterward, the models learn to reverse the diffusion process and reconstruct data samples from the induced noises. In response to the growing demand for multimedia generation, considerable research efforts have been devoted to

exploring text-to-image diffusion models. Stable Diffusion (SD)[40], rooted in the Latent Diffusion Models[34], significantly amplifies the scope of text-to-image synthesis applications.

With the input of Gaussian noises and user-elaborated prompts, SD generates diversified images through the denoising process. It has been demonstrated that the text prompts serve as an essential controller for image generation, with even minor modifications leading to significantly different outcomes. A body of works focuses on the role of image-text relationships in the generation, which is known as attention maps. For precise SD text-to-image generation, Prompt-to-Prompt[14] explores cross-attention layers during the diffusion backward inference process. Attend-and-Excite[5] modifies the latent codes by maximizing the dominant attention values for each subject token.

## 2.2 Diffusion based Customization

Despite the diversity of generated images by diffusion models, users often seek to synthesize specific concepts for their personal subjects. This task is named user customization[3, 22, 31, 47] and has been widely studied. LoRA[16] enables the diffusion models for specific styles or tasks by fine-tuning the extra parameters. This technique allows the model to associate image representations with the prompts describing them effectively. Textual Inversion[11] harnesses the diffusion models to reflect a novel concept to a pseudo-word, denoted as $S*$, within the malleable word embedding space of the text encoder. It allows for the seamless integration of unique concepts into the generative process, enhancing the model's ability to produce tailored and specific imagery.

DreamBooth[37] fine-tunes the diffusion model with a subject-identifier token consisting prompt (e.g., "a photo of *sks* person") and several images of the indicated subject. Through this process, the model becomes "familiar" with recognizing features associated with "*sks* person," enabling it to generate various representations in response to different prompts. Despite the convenience of diffusion-based customization, there exist some potential risks. Malicious users may misuse these tools to invade privacy, for instance, generating illicit pornographic images or harmful content. Such misuse highlights the importance of ethical considerations and robust safeguards in the development of text-to-image synthesis technologies.

## 2.3 Adversarial Attacks on Image Generation

Recognizing the urgency for privacy protection, researchers have introduced adversarial attacks to defend against malicious image manipulations. Inspired by adversarial attacks against classification tasks[1, 9, 27, 55], Yeh et al.[52] and Ruiz et al.[36] aim at distorting or neutralizing image editing DeepFake techniques. He et al.[13] harness the StyleGAN inversion technique, encoding the protected images into latent space and adding perturbations. CMUA-Watermark[17] and TAFIM[2] further train universal watermarks against various image manipulations.

Within the realm of diffusion-based customization for privacy protection, GLAZE[39] and AdvDM[23] employ invisible perturbations on personal images, protecting users from style theft and painting imitation. Differently, Anti-DreamBooth devises to generate adversarial noises and distort the outputs of customized diffusion methods, especially DreamBooth. SimAC[43] further takes

a sense of the perception in the frequency domain of images and leverages a greedy algorithm to select timesteps. Different from them, our DisDiff sets out from the image-text relationship and disrupts the internal textual guidance. Besides, we take into consideration the integration of diffusion sampling and adversarial learning and introduce the Merit Sampling Scheduler to adaptively restrict the perturbation updating amplitude.

## 3 Method

### 3.1 Preliminaries

**Stable Diffusion** consists of an autoencoder and a conditional UNet[35] denoiser. Firstly, the encoder $\mathcal{E}(\cdot)$ is devised to map a given image $x \in X$ into a spatial latent code $z = \mathcal{E}(x)$. The decoder $\mathcal{D}$ is then trained to map the latent code back to the input image such that $\mathcal{D}(\mathcal{E}(x)) \approx x$. Secondly, the conditional denoiser $\epsilon_\theta(\cdot)$ is performed to predict the added noises guided by the text prompt $y$. At this time, a pre-trained CLIP[30] text encoder $c(\cdot)$ is used to generate text embeddings. Representing the prompt $y$ as a conditioning vector denoted by $c(y)$, the diffusion model $\epsilon_\theta$ is trained to minimize the loss function:

$$L_{DM} = \mathbb{E}_{z \sim E(x), y, \epsilon \sim \mathcal{N}(0,\mathbf{I}), t} \| \epsilon - \epsilon_\theta(z_t, t, c(y)) \|_2^2 . \qquad (1)$$

During inference, latent $z_T$ is sampled from $\mathcal{N}(0, \mathbf{I})$ and progressively denoised to yield the latent $z_0$. Subsequently, $z_0$ is fed into the decoder to generate the image $x' = \mathcal{D}(z_0)$.

**DreamBooth** fine-tunes the diffusion model parameters for user customization. The training dataset consists of a subject-specific set $X_s$ and a class-specific set $X_c$, accompanied by corresponding prompts $y_s$ and $y_c$, e.g., "a photo of *sks* [class noun]" and "a photo of [class noun]", respectively. $X_s$ comprises various personal images, while $X_c$ serves to mitigate model overfitting. Therefore, Dream-Booth utilizes a two-part loss to train the diffusion models:

$$L_{DB} = \mathbb{E}_{z \sim E(x_s), \epsilon \sim N(0,\mathbf{I}), t} \| \epsilon - \epsilon_\theta(z_t, t, c(y_s)) \|_2^2$$
$$+ \lambda_{DB} \mathbb{E}_{z \sim E(x_c), \epsilon \sim N(0,\mathbf{I}), t} \| \epsilon - \epsilon_\theta(z_t, t, c(y_c)) \|_2^2, \qquad (2)$$

where $\lambda_{DB}$ represents a balanced hyper-parameter. In the inference process, users modify the subject-specific prompt to their desire, e.g., "a photo of *sks* [class noun] on the mountain". The model then generates this specific subject on the mountain as described.

**Adversarial attacks** manipulate model $f$ by introducing imperceptible perturbations to the input data. As to the image generation, the attacker aims to distort or nullify the generation model's outputs by an optimal perturbation $\delta$. The perturbations are bounded within an $\eta$-ball according to distance metric $\ell_p$, where $\eta$ represents the maximum allowed perturbation magnitude. The goal of adversarial attacks is to maximize the loss function, ensuring the outputs are different from the ground truth $y_{true}$. $\delta_{adv}$ is optimized by:

$$\delta_{adv} = \arg \max L(f(x + \delta), y_{true}), s.t. \|\delta\|_p \leq \eta. \qquad (3)$$

Projected Gradient Descent (PGD)[27] is a widely used method for adversarial attacks. The perturbations are computed by disrupting the input along the direction of Eq.3 iteratively. Every iteration process of PGD updates the adversarial example $x'$ as follows:

$$x'_0 = x, \qquad (4)$$

$$x'_k = \Pi_{x,\eta}(x'_{k-1} + \gamma \cdot \text{sgn}(\nabla_x L(f(x'_{k-1}), y_{true}))), \qquad (5)$$

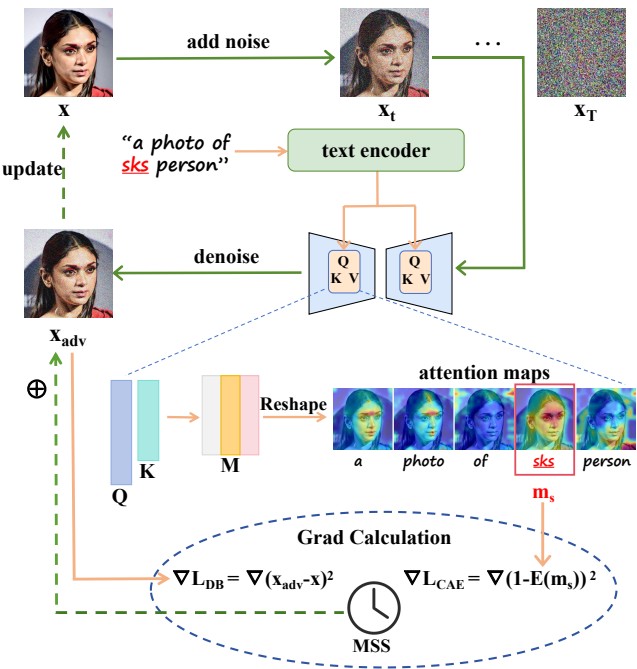

**Figure 2: The overview framework of Disrupting Diffusion. The prompt "a photo of *sks* person" is used for training. At every denoising process, we acquire the attention maps of the subject to calculate $L_{DB}$ and $L_{CAE}$. Then, the gradients are aggregated and fed into the Merit Sampling Scheduler, which is used for PGD attack to update $x_{adv}$.**

where $\Pi_{x,\eta}(z)$ confines the pixel values of $z$ within the $\eta$-ball, $\gamma$ represents the step size, and $k$ denotes the number of iterations.

## 3.2 Cross-Attention Erasure

**Motivation.** As in the literature[5, 14], cross-attention maps represent the influence of tokens on image pixels, which control the text-guided image generation in the diffusion model. Thus, we first delve into the training process of DreamBooth through the lens of attention maps. Specifically, given a textual guidance composed of sequence $W = \{w_1, w_2, ..., w_n\}$, we derive the corresponding attention maps $M = \{m_1, m_2, ..., m_n\}$, where n is the word number. At timestep $t$, $m_t$ is computed through the diffusion forward process with latent code $z_t$:

$$m_t = \text{Softmax}\left(\frac{QK^\top}{\sqrt{d}}\right), \qquad (6)$$

where query $Q$ is equal to $W_Q \cdot c(y)$, key $K$ is equal to $W_K \cdot c(y)$. $W_Q$ and $W_K$ are weight parameters of the query and key projection layers and $d$ is the channel dimension of key and query features.

Notably, we aggregate these attention maps by averaging across all $16 \times 16$ attention layers and heads, as they are proved to convey the most semantic information[14]. Besides, it is observed that the original attention maps may not fully reflect whether an object is generated in the image[5]. In other words, a single patch with high attention value might originate from partial information passed

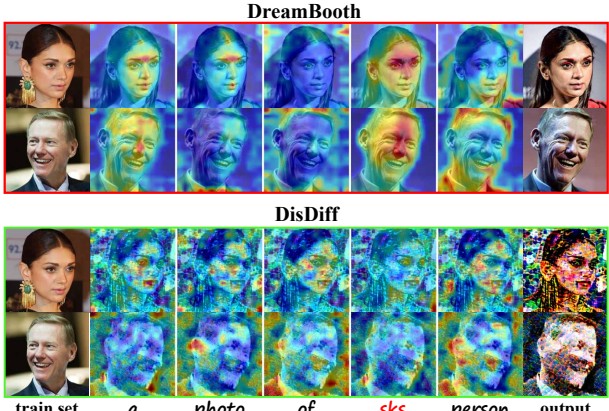

**Figure 3: Visualizations of cross-attention maps from unprotected DreamBooth and DisDiff. For the unprotected ones, the diffusion model captures the subject-identifier token "*sks*" (highlighted red areas on the face) and generates customized images. However, DisDiff erases the model's attention on that token and dramatically distorts the model's outputs.**

from the token. To mitigate this, we further apply a Gaussian filter over $m_t$, which ensures that the attention value of the maximally-activated patch depends on its neighboring patches, facilitating a smoother attention distribution across the image.

Figure 3 shows the visualizations of attention maps. As the customization prompt is characterized by a subject-identifier token $w_s$("*sks*") and a class-identifier token $w_c$("*person*"), we maintain our focus on their related attention maps. The red highlighted areas reveal high relativity between the token and the generated image. Clearly, for unprotected subjects, the fine-tuned diffusion model successfully associates the subject-identifier token "*sks*" with faces. However, the class-identifier token "*person*" becomes less prominent. That is, "*sks*" actually stands on the subject. We compute the relative energy of the subject-identifier token by:

$$E(m_s) = \frac{\sum_{A \in m_s} A^2}{\sum_{m_i \in M} \sum_{A \in m_i} A^2}, \qquad (7)$$

where $m_s$ is the attention map of $w_s$ and $A$ stands on the pixel value in the maps. $E(m_s)$ reflects the energy weight percentage of the subject-identifier token. To protect personal privacy from customization, we thereby try to *"erase"* the energy associated with the values in $m_s$. Concretely, the proposed Cross-Attention Erasure loss is as follows:

$$L_{CAE} = (1 - E(m_s))^2. \qquad (8)$$

Our adversarial attack focuses on maximizing $L_{cae}$, and the optimal perturbation $\delta$ can be learned by training procedure in Section 3.4.

The intuition of our CAE module is that after erasing the cross-attention map of the subject-identifier token (with eliminated highlighted areas), the model would lose the direction of text guidance. As a result, the generated images would exclude the identity information or even the subjects themselves. The protecting results are shown in the last two rows in Figure 3. Compared with highlighted red areas of unprotected DreamBooth(the first two rows), the impact

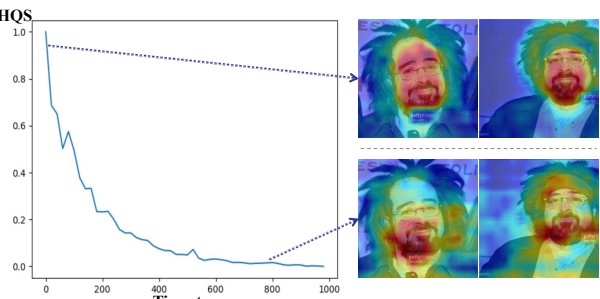

**Figure 4: Analysis of varying timesteps. As the timestep becomes larger, not only does the HQS metric significantly decrease but also the model pays more attention to identity-irrelevant information, illustrating that the steps in the former are more important for perturbation training.**

of the subject-identifier token on image generation dramatically decreases. In light of this, the learned perturbations distort image-text correlations of the diffusion model, affecting unidentified or unrecognized inference outputs.

### 3.3 Merit Sampling Scheduler

Note that in the adversarial learning procedure of DisDiff, the timestep sampling process of diffusion models is included. Thus, to evaluate the importance of steps for adversarial learning, we adopt the Hybrid Quality Score (HQS) metric inspired by [44]. Assuming that $G_t$ is the computed gradients on images, we first calculate the L1 norm of the gradients by:

$$N_t = \sum_i |G_t^i|. \tag{9}$$

$G_t$ is then converted by a softmax function to the confident map $p_t$. The entropy is computed by:

$$H_t = -\sum_i p_t^i \log p_t^i. \tag{10}$$

Afterward, the sequences $N = \{N_1, N_2, ..., N_T\}$ and $H = \{H_1, H_2, ..., H_T\}$ are normalized to [0,1] and the HQS metric is computed as:

$$\text{HQS}(t) = \text{norm}(N_t) - \text{norm}(H_t). \tag{11}$$

Figure 4 shows the HQS evaluation of various timesteps. High HQS stands on high values of gradient norm and low entropy, indicating that the model pays more attention at this timestep and mainly focuses on identity-related areas (such as faces). During early iterations when HQS is high, larger PGD steps may be beneficial for adversarial attacks. Conversely, as the HQS decreases, smaller PGD steps can help fine-tune the generated adversarial samples, as well as avoid excessive deviation from the original input distribution.

Besides, we also visualize the cross-attention map of the subject-identifier token for different timesteps. As can be seen, when the timesteps are large, the model pays more attention to the irrelevant information of the identity. As the timestep gets smaller, the attention gradually moves to the identity's faces. Observations on HQS and attention maps both empirically justify that the timesteps sampling is essential to adversarial attacks.

---

**Algorithm 1** Disrupting Diffusion Step

1: **Input:** Stable Diffusion model $SD$, protected image $X_{adv}$, clean images batch $X_{cl}$, prompt $W = \{w_1, w_2, ..., w_n\}$, scheduler $h(t)$, surrogate model steps $t_1$, PGD steps $t_2$
2: **procedure** $DBSTEP(SD, X, W)$
3:     Random sample timestep $t$
4:     $X', \_ \leftarrow SD(X, W, t)$
5:     $L_{DB} \leftarrow MSE(X'_s, X_s)$
6:     $SD \leftarrow SD - \alpha_{DB} \nabla L_{DB}$
7:     **return** $SD$
8: **end procedure**
9: **For** $t$ in $\{1, 2, ..., t_1\}$ :
10:     $SD \leftarrow DBSTEP(SD, X_{cl}, W)$
11: **For** $t$ in $\{1, 2, ..., t_2\}$ :
12:     $X'_{adv}, M_t \leftarrow SD(X_{adv}, t, W)$
13:     $M_t \leftarrow \text{Softmax}(M_t)$
14:     $L_{DisDiff} \leftarrow MSE(X'_{adv}, X_{adv}) + \lambda \cdot L_{CAE}(M_t)$
15:     $X_{adv} \leftarrow X_{adv} + \gamma \cdot h(t) \cdot \text{sgn}(\nabla_{X_{adv}} L_{DisDiff})$
16: **For** $t$ in $\{1, 2, ..., t_1\}$ :
17:     $SD \leftarrow DBSTEP(SD, X_{adv}, W)$

---

Therefore, we resort to adopting a time-dependent function as the Merit Sampling Scheduler (MSS) to adjust the perturbation updating steps in PGD adaptively. We empirically design the time-aware decreasing function as:

$$h(t) = \frac{1}{2}(cos(\frac{\pi t}{T}) + 1), \tag{12}$$

where $T$ is the max timestep to sample so that the amplitude of $h(t)$ is limited in [0,1]. In this regard, our proposed MSS dynamically adjusts the step size of the perturbation updating. The scheduler is then used to be an enhancement for PGD attacks. Given the overall loss $L$, PGD process in Eq.5 is modified as:

$$x'_k = \Pi_{x,\eta}(x'_{k-1} + \gamma \cdot h(t) \cdot \text{sgn}(\nabla_x L(f(x'_{k-1}), y_{\text{true}}))). \tag{13}$$

By incorporating MSS, we associate adversarial learning with the diffusion model sampling process. The modified PGD updates the perturbations precisely, contributing to better attack performances.

### 3.4 Training Procedure

To sum up, the proposed DisDiff combines the DreamBooth loss with the Cross-Attention Erasure loss as the overall loss:

$$L_{DisDiff} = L_{DB} + \lambda L_{CAE}, \tag{14}$$

where $\lambda$ is a trade-off hyper-parameter. To conduct the adversarial attack, we utilize modified PGD (Eq.13) and maximize the loss function to train the perturbations:

$$\delta = \underset{\delta}{\text{argmax}}\, L_{DisDiff}. \tag{15}$$

We conduct an alternative training strategy similar to [42]: at each epoch, we first adopt a clean image set $X_c$ to train the surrogate model for $t_1$ steps. The adversarial example $x'$ is then updated by PGD attack for $t_2$ steps and the model is once again trained by the adversarial examples. The training process of DisDiff is illustrated in Algorithm 1. For convenience, we omit the class-specific datasets and prompts of the DreamBooth procedure.

**Table 1: The comparison of attack performances on VGGFace2 and CelebA-HQ datasets.**

| Dataset | Method | "a photo of sks person" | | | | "a dslr portrait of sks person" | | | |
|---|---|---|---|---|---|---|---|---|---|
| | | FDFR↑ | ISM↓ | FID↑ | BRISQUE↑ | FDFR↑ | ISM↓ | FID↑ | BRISQUE↑ |
| VGGFace2 | w/o Protect[37] | 0.06 | 0.56 | 236.37 | 20.37 | 0.21 | 0.44 | 279.05 | 7.61 |
| | AdvDM[23] | 0.10 | 0.38 | 359.65 | 17.39 | 0.11 | 0.29 | 397.64 | 27.97 |
| | Anti-DB[42] | 0.62 | 0.32 | 462.12 | 37.10 | 0.72 | 0.27 | 448.98 | 38.89 |
| | **DisDiff** | **0.77** | **0.27** | **476.28** | **42.46** | **0.95** | **0.06** | **473.38** | **41.26** |
| CelebA-HQ | w/o Protect[37] | 0.07 | 0.63 | 154.63 | 13.15 | 0.30 | 0.46 | 221.89 | 8.75 |
| | AdvDM[23] | 0.06 | 0.59 | 197.59 | 24.55 | 0.30 | 0.42 | 235.47 | 16.88 |
| | Anti-DB[42] | 0.56 | 0.44 | 386.06 | 41.58 | 0.48 | **0.33** | 384.80 | 35.86 |
| | **DisDiff** | **0.62** | **0.40** | **412.19** | **44.98** | **0.55** | 0.34 | **400.38** | **38.35** |

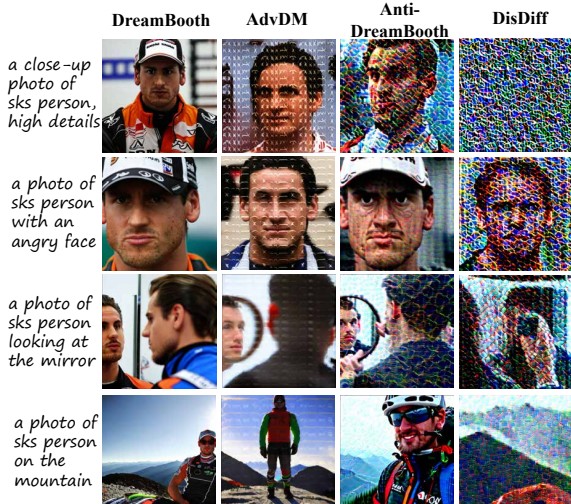

**Figure 5: Comparison under other inference prompts. Four rows show different image edit prompts: distance, expression, action, and location, respectively.**

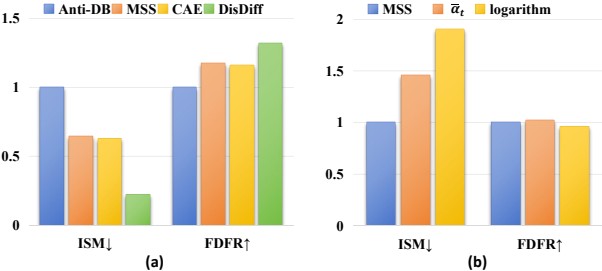

**Figure 6: Ablation study of (a) the proposed module CAE and MSS, (b) different schedulers.**

# 4 EXPERIMENTS

## 4.1 Experimental Setup

**Evaluation Benchmarks.** We quantitatively evaluate our approach on two benchmarks: CelebA-HQ[26] and VGGFace2[4]. For each dataset, we choose 50 identities as the protecting subjects. 8 different images are prepared for each subject and divided equally into two subsets: the clean image set and the target protection set. All images undergo center-cropping and resizing procedures, resulting in a uniform resolution of 512 × 512.

**DreamBooth and Adversarial Attack Procedures.** We address three versions of pre-trained SD from HuggingFace[10]: v1.4, v1.5 and v2.1. For DreamBooth, the text-encoder and UNet model are trained with a batch size of 2 and the learning rate of $5 \times 10^{-7}$ for 1,000 training steps. The prompt is "a photo of *sks* person" for the subject images and "a photo of person" for class-specific datasets. The PGD attack is configured with $\gamma$ = 0.005 and a default noise budget $\eta$ = 0.05 for VGGFace2 and 0.07 for CelebA-HQ respectively. We train the perturbations by 50 iterations, each containing 3 steps

of surrogate model training and 6 steps of perturbation training. The hyper-parameter $\lambda$ in $L_{DisDiff}$ is set to 0.1 for both datasets.

**Evaluation Metrics.** We utilize two widely used image quality metrics, FID[15] and BRISQUE[28], to test the disrupting results. Besides, images generated from these models may lack detectable faces, which we quantify as the Face Detection Failure Rate (FDFR), measured by the RetinaFace detector[7]. We also extract face recognition embeddings using the ArcFace recognizer[8] and compute the cosine distance to the average face embedding of the entire user's clean image set, referred to as Identity Score Matching (ISM).

## 4.2 Comparison with SOTAs

We conduct comparisons with two SOTA anti-customization baselines: AdvDM[23] and Anti-DreamBooth[42]. For a fair comparison, we reproduce their source codes and train adversarial examples with equivalent noise budgets. These protected examples are then used to fine-tune the diffusion models (DreamBooth), with prompt "a photo of *sks* person." Then, the trained diffusion models are tested by prompts "a photo of *sks* person" and "a dslr portrait of *sks* person". As can be seen in Table 1, our method demonstrates notable performances across four metrics and two prompts. Specifically, higher FDFR rates signify a decreasing number of detected faces, while lower ISM rates indicate less similarity with the original subject. Increased FID and BRISQUE rates emphasize the lower image quality, resulting in significant distortions of the outputs.

To further verify the efficacy, we also test our method under various prompts with distance, expression, action, and location descriptions, respectively. As shown in Figure 5, DreamBooth edits

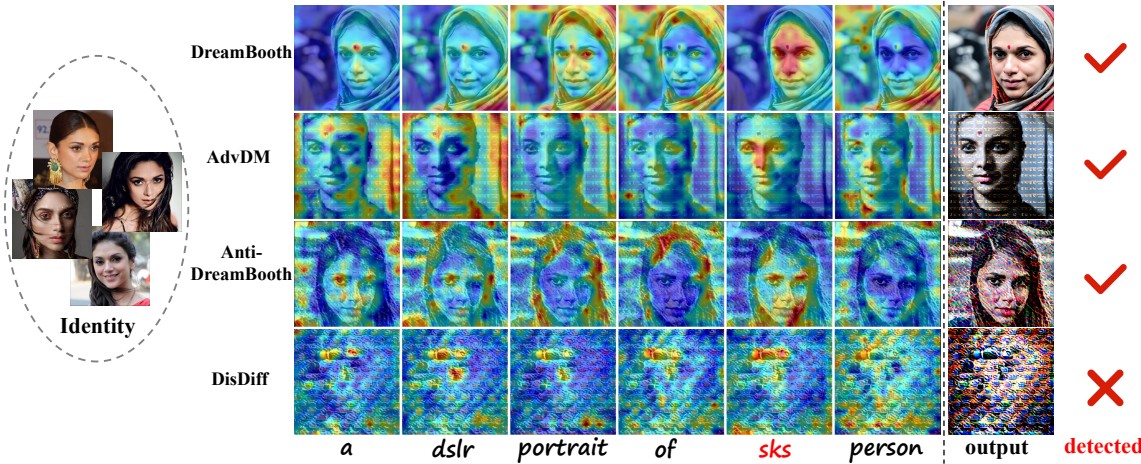

**Figure 7: Visualizations of identity images, attention maps, and generated images before and after protecting.**

the images as the prompts describe. AdvDM pushes the outputs to the target image so that the results show a similar texture to that image. Although Anti-DreamBooth distorts the output images, the identity is still recognizable. Our proposed DisDiff leads to more unrecognized faces and unidentified subjects by directly disrupting the cross-attention module, which greatly boosts the performances.

## 4.3    Ablation Study

**Impact of Cross-Attention Erasure.** To testify to the impacts of our proposed Cross-Attention Erasure, we visualize the attention maps and the corresponding generated images. As shown in the first row of Figure 7, highlighted red areas occur on the identity face for the token "*sks*", which indicates that the diffusion model pays much more attention to the learned identity with "*sks*". This is intuitive since DreamBooth progressively makes the token "*sks*" and the special person connected closely. This situation still exists when the identity is protected by AdvDM and Anti-DreamBooth, which is undesired for privacy protection. Compared with them, DisDiff successfully erases the impact of "*sks*". That is, the diffusion model ignores the token "*sks*" and its guidance on generation, thus has no idea about whom to generate. The images protected with our method thus exhibit greater blurriness and indistinctiveness.

**Module and Scheduler Selection.** We conduct the ablation study of the proposed Cross-Attention Erasure (CAE) and Merit Sampling Scheduler (MSS) modules. As shown in Figure 6(a), we set the metrics of baseline method Anti-DreamBooth as 1 and calculate the relative magnitude of DisDiff. Clearly, when utilizing CAE or MSS, the lower ISM and higher FDFR illustrate that the customization outputs become unidentified and unrecognized, indicating DisDiff's effectiveness. Also, we apply two widely used decreasing functions to serve as the scheduler in MSS: the logarithm decreasing function[1] and diffusion models sampling sequence $\bar{\alpha}_t$. We set the scheduler in Eq.12 as the baseline and compare the attack performances in Figure 6(b). Finally, we select the cosine function in Eq.12 as the default, which achieves the lowest ISM rate.

---

[1]The logarithm decreasing function is $1 - \frac{log(t+1)}{log(T+1)}$, where T is the max sampling step.

**Table 2: Attack performances with different noise budget $\eta$ and * indicates the default setting.**

| $\eta$ | LPIPS | FDFR↑ | ISM↓ | FID↑ | BRISQUE↑ |
|---|---|---|---|---|---|
| 0 | - | 0.06 | 0.56 | 236.37 | 20.37 |
| 0.01 | ≈ 0 | 0.08 | 0.53 | 271.92 | 33.93 |
| 0.03 | 0.01 | 0.59 | 0.37 | 426.68 | 40.20 |
| 0.05∗ | 0.03 | 0.77 | 0.27 | 476.28 | 42.46 |
| 0.07 | 0.07 | 0.82 | 0.26 | 485.02 | 43.63 |
| 0.09 | 0.11 | 0.85 | 0.16 | 494.51 | 40.83 |

**Noise Budget.** The noise budget $\eta$ represents the permissible magnitude of perturbations, and a large noise budget is easily perceptible by humans. LPIPS is used to measure the image quality before and after attack. As shown in Table 2, DisDiff shows high FID and BRISQUE scores when the budget is as small as 0.03. Besides, LPIPS is still low when we set $\eta = 0.05$(default), meaning that the perturbation is still invisible. Moreover, the attack performances become stronger as the budget increases, which is also reasonable since the larger the perturbation is, the greater the attack performances can be achieved.

## 4.4    Robust Analysis

**Attacks on different generators.** Considering the potential malicious uses of DreamBooth across different diffusion model generators, we conduct experiments to assess the impact. As shown in Table 3, we first test under the self-surrogate setting (indicated "self"), where the perturbation training and DreamBooth models are the same version. Experiments show that DisDiff achieves better performances than Anti-DreamBooth and generalizes well on SD v1.4 and v1.5. Then, we test the generated perturbations robustness under different generators (indicated "v2.1"). We employ the adversarial examples generated with SD v2.1 to train DreamBooth under v1.4 and v1.5. The FDFR approaches 1.0 and ISM approaches

**Table 3: Attack performances on different generator versions on VGGFace2. "Self" means the perturbation training and DreamBooth process share the same surrogate model, while "SD v2.1" means the perturbation training with SD v2.1 and DreamBooth with other versions.**

| Surrogate | Method | SD v1.4 | | | | SD v1.5 | | | |
|---|---|---|---|---|---|---|---|---|---|
| | | FDFR↑ | ISM↓ | FID↑ | BRISQUE↑ | FDFR↑ | ISM↓ | FID↑ | BRISQUE↑ |
| w/o Protect | | 0.09 | 0.56 | 231.13 | 19.79 | 0.09 | 0.55 | 217.22 | 18.33 |
| self | Anti-DB | 0.88 | 0.07 | 511.59 | 42.77 | 0.93 | 0.07 | **517.03** | 46.09 |
| | **DisDiff** | **0.96** | **0.04** | **522.57** | **46.26** | **0.99** | **0.05** | 512.70 | **55.90** |
| v2.1 | Anti-DB | 0.89 | 0.10 | 495.50 | 38.25 | 0.90 | 0.12 | 506.64 | 44.02 |
| | **DisDiff** | **0.99** | **0.02** | **505.66** | **44.60** | **0.96** | **0.08** | **510.93** | **50.83** |

**Table 4: Attack performances on VGGFace2 when the DreamBooth training prompt and subject-identifier token are different from the perturbation training stage. $S_*$ is "t@t" for the first row and "sks" for the second row.**

| DreamBooth prompt | Method | "a photo of $S_*$ person" | | | | "a dslr portrait of $S_*$ person" | | | |
|---|---|---|---|---|---|---|---|---|---|
| | | FDFR↑ | ISM↓ | FID↑ | BRISQUE↑ | FDFR↑ | ISM↓ | FID↑ | BRISQUE↑ |
| "a dslr portrait of sks person" | Anti-DB | 0.01 | **0.16** | **279.37** | 19.74 | 0.57 | 0.30 | 459.65 | 39.25 |
| | **DisDiff** | **0.05** | 0.20 | 270.47 | **21.95** | **0.85** | **0.21** | **476.27** | **40.14** |
| "sks" → "t@t" | Anti-DB | 0.60 | 0.34 | 454.53 | 40.61 | 0.41 | 0.31 | 396.92 | 34.50 |
| | **DisDiff** | **0.73** | **0.28** | **464.09** | **42.24** | **0.71** | **0.25** | **414.26** | **38.87** |

0, indicating minimal faces generated. These results illustrate that the adversarial examples perform well against different generators.

**Attacks while prompt or subject-identifier mismatching.** Assuming that the adversarial learning prompts differ from the ones used to train DreamBooth, we conduct experiments to test the robustness, as illustrated in Table 4. In the first row, we employ "a photo of *sks* person" for adversarial learning and "a photo of a dslr portrait of *sks* person" for DreamBooth, conducting inference for both. DisDiff and Anti-DreamBooth perform poorly when confronted with the prompt "a photo of *sks* person" due to prompt mismatching. However, when presented with the prompt "a dslr portrait of *sks* person", DisDiff demonstrates superior performances. This is partially due to that the training prompt becomes more complicated and the models are overfitting.

We also evaluate the results by changing the DreamBooth subject-identifier token to "*t@t*", as in the second row. Our DisDiff remains well-generalized to "*t@t*". The reason is that we apply the softmax function in CAE. The summary of all the softmax attention maps for every pixel is 1. After erasing the attention map of "*sks*", the values of other tokens become larger. As a result, the attention map of substitute token "*t@t*" is erased even though it is not involved in perturbation training. To sum up, DisDiff is robust to different DreamBooth prompts and subject-identifier tokens.

**Attacks on different customization methods.** To verify our protection generality across various generation methods, we reproduce wildly used customization methods: Low-rank Adaptation (LoRA)[16] and Textual Inversion (TI)[11]. We train customization with LoRA and TI and conduct adversarial attacks with Anti-DreamBooth and DisDiff. As can be seen in Table 5, compared to unprotected DreamBooth, both Anti-DreamBooth and DisDiff successfully attack these two customization methods. DisDiff shows

**Table 5: Attack performances on other diffusion customization methods on VGGFace2.**

| | Method | FDFR↑ | ISM↓ | FID↑ | BRISQUE↑ |
|---|---|---|---|---|---|
| LoRA | w/o Protect | 0.11 | 0.43 | 248.05 | 17.31 |
| | Anti-DB | 0.68 | 0.36 | 403.38 | 44.38 |
| | **DisDiff** | **0.73** | **0.21** | **418.70** | **44.70** |
| TI | w/o Protect | 0.04 | 0.42 | 222.70 | 7.59 |
| | Anti-DB | 0.09 | 0.27 | 289.66 | 39.37 |
| | **DisDiff** | **0.14** | **0.24** | **308.77** | **41.23** |

better evaluation scores, indicating the robustness of protection ability across different customization methods.

## 5 Conclusion

In this paper, we propose a novel adversarial attack method, Disrupting Diffusion (DisDiff). We track the cross-attention maps and propose a Cross-Attention Erasure module to disrupt these maps. We also introduce a time-aware Merit Sampling Scheduler to adaptively adjust the PGD steps. DisDiff successfully disrupts customization's outputs as unrecognizable and unidentifiable.

## Acknowledgments

This work was supported by the National Key R&D Program of China under Grant 2022YFB3103500, the National Natural Science Foundation of China under Grants 62202459 and 62106258, and Young Elite Scientists Sponsorship Program by CAST (2023QNRC001) and BAST (NO.BYESS2023304), and Beijing Natural Science Foundation QY23179.

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
