# OpenReview forum: "Disrupting Diffusion: Token-Level Attention Erasure Attack against Diffusion-based Customization"
_acmmm.org/ACMMM/2024/Conference — MM2024 Poster_

### Official Review · Reviewer_mBLr · 2024-05-23

**Rating:** 4
**Confidence:** 3

**Summary:**

This paper proposes an adversarial attack against diffusion-based customization, to protect human privacy. The proposed attack disrupts the mapping relationship between the trigger word and the input images through a Cross-Attention Erasure module. In addition, they adopt a Merit Sampling Scheduler to restrict the perturbation updating step length in a step-aware manner. Finally, the proposed method achieves excellent performance compared with Sota methods.

**Strengths:**

- S1: This paper conducts comprehensive experiments and the proposed method is superior to Sota methods.

- S2: This paper has a clear logic and shows reasonable explanations for the erasure module and perturbation updation strategy.

**Limitations:**

- W1: The authors would better consider "Uncontrolled Settings" which have been considered in Anti-DreamBooth. I am curious about the effectiveness of the proposed method when the input images consist of both clean images and adversarial images.

- W2: The authors evaluate the proposed attack on DreamBooth, Low-rank Adaptation (LoRA), and Textual Inversion (TI). However, these methods need much time for tuning and many tuning-free customization approaches, such as PhotoMaker (https://arxiv.org/abs/2312.04461) and IPAdapter (https://arxiv.org/abs/2308.06721), have been proposed. Therefore, the author would better evaluate the proposed method on the Sota customization methods.

- W3: Although this paper analyzes the noise budget in the ablation study, it still lacks the visualization results of the adversarial images. Some noises are visible to humans, and image quality metrics such as LPIPS cannot effectively reflect them.

- W4: Finally, there are several small questions:

  - In line 564, the noise budgets for VGGFace2 and CelebA-HQ are set as 0.05 and 0.07, respectively. I notice that the two budgets in Anti-DreamBooth are both set as 0.05. What is the basis for choosing different thresholds?
  - In Table 1, it seems unnecessary to display a reference column. By the way, the authors would like to update the reference information in the Reference Section from Arxiv versions to published versions.
  - In lines 790-791, if you don't provide the results of Custom Diffusion in Table 3, it seems unnecessary to mention it here.

**Suitability:**

3

---

### Official Review · Reviewer_RWTp · 2024-05-23

**Rating:** 4
**Confidence:** 3

**Summary:**

This paper proposes Disrupting Diffusion (DisDiff), an effective new approach to proactively disrupt facial image customization and protect user privacy by erasing key attention guidance and adapting to the diffusion sampling process during the adversarial attack.

**Strengths:**

1. DisDiff introduces the Cross-Attention Erasure (CAE) module, which directly targets and "erases" the critical attention maps that guide the diffusion model to generate the customized subject. By disrupting this key mechanism, DisDiff effectively prevents the model from generating identifiable images of the protected individual.
2. The Merit Sampling Scheduler (MSS) analyzes the importance of each timestep in the diffusion sampling process and adaptively adjusts the adversarial attack's perturbation update step size. This allows DisDiff to more effectively disrupt the model by focusing the attack on the most impactful timesteps.
3. Experiments demonstrate that DisDiff outperforms state-of-the-art adversarial attack methods, achieving a 12.75% higher face detection failure rate (FDFR) and 7.25% lower identity score matching (ISM) on average. DisDiff is also shown to be robust across various customization methods.

**Limitations:**

1. Does DisDiff generalize to objects other than malicious customizations of facial images?
2. While the paper does test robustness to Gaussian blur and JPEG compression, there are many other common pre-processing steps that could impact the effectiveness of the adversarial attacks, such as resizing, cropping, color space transformations, etc. A more thorough evaluation of robustness to a wider range of pre-processing would strengthen the results.
3. The paper sometimes uses the abbreviation "DB" for "DreamBooth" without first defining it, e.g. in Table 3 and a few other places. It would be better to define this abbreviation on first use.

**Suitability:**

3

---

### Official Review · Reviewer_2npb · 2024-05-25

**Rating:** 4
**Confidence:** 3

**Summary:**

This paper delves into the image-text relationships in diffusion models that find the subject-identifier tokens are easily disturbed.
Then, the authors propose a cross-attention erasure module to explicitly ``erase’’ privacy information, achieving better protection performance than Anti-DreamBooth.

**Strengths:**

1.  The motivation of the cross-attention erasure module is well delivered. Visualization of the cross-attention map is very helpful.
2. A Merit Sampling Scheduler is proposed to guide the perturbation updating step length in PGD.
3. The paper is clearly written and organized. Extensive experiments are conducted to evaluate the effectiveness of DisDiff.

**Limitations:**

1. Visualization comparison between the original image and protected counterpart is not shown.
2. The robustness of DisDiff is poor, as shown in Table 6. Gaussian Blur and JPEG Compression can easily corrupt its effectiveness. More image distortions are not discussed. Additionally, what's the meaning of "No attack, no preproc."? It's not illustrated in the main paper.
3.  In Table 4, why did training while testing on SD v1,4 achieve better performance than on training and testing on the same model (Self)?
4. Efficiency of DisDiff should be considered.

**Suitability:**

3

---

### Official Review · Reviewer_PKms · 2024-05-27

**Rating:** 4
**Confidence:** 3

**Summary:**

The paper presents a novel method to solve the privacy leakage problem caused by customization based on diffusion model, \emph{i.e.,} DisDiff, a proactive adversarial attack method for privacy preservation. The authors also explore the image-text relationship to justify its importance on generation guidance, based on which a novel cross-attention erasure module to eliminate the subject-identifier token-related attention map. Besides, a merit sampling scheduler is introduced to restrict the perturbation updating step length in a step-aware manner.

**Strengths:**

1.	The idea of solving privacy issue in diffusion-based customization by cross-attention is novel.
2.	The Experiments indicate the superiority of the work.

**Limitations:**

1.	Figure 1 well indicates the superiority of the proposed DisDiff to Anti-DB and the motivations of the DisDiff. However, it may not well illustrate the target application case.
2.	DisDiff includes two modules, and the relationship between the two modules should be specified.
3.	It would be better to analyze the influence of cross attention erasure on the diffusion theoretically, if possible.

**Suitability:**

3

---

### Meta-Review · Area_Chair_u9Pa · 2024-07-01

**Recommendation:** Accept (Poster)
**Confidence:** 2

**Metareview:**

Summary:

This paper presents DisDiff, a proactive adversarial attack method aimed at preserving privacy in diffusion-based image customization. It introduces a Cross-Attention Erasure (CAE) module and a Merit Sampling Scheduler (MSS) to disrupt the generation of identifiable images and improve privacy protection.

Strengths:
1. The paper introduces a novel method named DisDiff, which addresses privacy leakage in diffusion-based customization.
2. Extensive experiments demonstrate DisDiff's superiority over state-of-the-art techniques, with improvements in privacy protection metrics.
3. The paper is well-written and organized, with clear visualizations that enhance the understanding of the proposed method.

Limitations:
1. The relationship between the CAE and MSS modules needs clearer explanation, and the theoretical impact of cross-attention erasure on diffusion requires further analysis.
2. The robustness of DisDiff against various image distortions and in uncontrolled settings needs more thorough evaluation.
3. Visualization comparisons and some practical considerations are lacking in the paper such as the explanation of noise budgets.

I will temporarily recommend this paper as Accept (Poster) without complete confidence. This case is very special, where all reviewers had positive initial ratings, while two of them changed their decision to Borderline Reject after reading the rebuttal. Therefore, I sincerely consider this case needs to be discussed.